# Mixing, Fast and Slow: Assessing the Efficiency of Electronically Conductive Networks in Hard Carbon Anodes

Manisha Anne Sawhney [1,2,*] and Jenny Baker [1,2]

1 Faculty of Science and Engineering, Swansea University Bay Campus, Fabian Way, Swansea SA1 8EN, UK
2 SPECIFIC Innovation and Knowledge Centre, Swansea University Bay Campus, Fabian Way, Swansea SA1 8EN, UK
* Correspondence: m.a.sawhney@swansea.ac.uk

**Abstract:** This work aimed to answer fundamental questions about the optimal processing and formulation of hard carbon electrodes typical of those anticipated in commercial sodium-ion cells. Procedurally simple tests were proposed to compare the effects of slurry mixing energy and conductive additives on the morphology of and conductive networks in electrodes made with hard carbons from two different manufacturers. Long-range and short-range electronic conductivity was quantified with high repeatability for samples of each hard carbon electrode produced on different days. The most significant changes induced by mixing energy were observed in the electrodes produced without conductive additives, which was found to relate to post-processing particle size. Hard carbon from one source was pulverized by high energy mixing, replacing the electronic effect of conductive additives while increasing pore tortuosity and impedance. These findings recommend evaluating the dry electrode through-resistance as a complement to quantifying pre-cycling impedance to validate mixing protocol and the application of conductive additives in hard carbon electrodes. These procedures can also serve as reliable low-cost methods for quality control at early stages of sodium-ion anode manufacturing.

**Keywords:** Na-ion; contact resistance; impedance modulus; electrode microporosity

## 1. Introduction

The advancement of sodium-ion (Na-ion) batteries as more cost-effective and sustainable alternatives to lithium-ion batteries has been hindered by both practical and fundamental issues [1]. While gravimetric capacity is less critical for applications in stationary energy storage installations, low initial coulometric efficiency and lifetime capacity retention remain critical barriers to the competitiveness of Na-ion technology [2]. Design disadvantages particular to Na-ion development primarily originate in mechanisms at the anode surface, where unstable electrochemical phenomena compromise the reversibility of the sodium intercalation [3]. Despite the widespread substitution of graphite with hard carbon as an anode active material to improve the sodiation efficiency in Na-ion cells, the optimization of this anode–electrolyte interface remains elusive [4].

Much of the current research into improving hard carbon anode performance prescribes novel precursor feedstocks [5,6], though reported tests are rarely performed on a commercial hard carbon for comparison with a control. Another common approach to anode enhancement involves chemical modifications [7,8], the adoption of which would increase manufacturing complexity and costs. Electrochemically inactive components of anodes, such as polymeric binders and conductive additives, are often used at high proportions (10% each) in sodium-ion anode research, while less study is dedicated to optimizing and formulating these aspects of electrodes [9]. The use of these materials has been adopted directly from practices common in lithium-ion cell manufacturing; this includes adding conductive carbon black, which is produced through energy-intensive processes using fossil fuel feedstocks [10].

Carbon black is blended into hard carbon electrode slurries not for its electrochemical activity but to enhance the electronic connections of the active material particles to the current collector, which allow for low-impedance current pathways between the electrochemically active interface and the external circuitry [11]. Electronic resistance caused by poor connections between particles (short-range) across electrode thickness (long-range) or between the film and current collector (contact resistance) contribute to a higher cell internal resistance, which consumes energy during each charge or discharge. Since electrochemical phenomena contribute proportionally more to internal resistance than electronic inefficiencies, traditional voltage vs. current methods are not suitable for distinguishing between ionic and electronic sources of resistance [12]. Galvanostatic cycling fixes both voltage and current parameters; therefore, this method cannot describe the internal resistance of a cell, a value sometimes estimated based on assumptive models [13].

In a composite electrode containing non-conductive components, the quantitative relation between the conductive additive and the electronic conductivity is called the percolation curve [14,15]. Graphite inherently possesses electrical conductivity, but commercial electrode formulas apply conductive additives [16] since the anisotropic orientation of stacked sheets and the irregular spacing between the particulates cause discontinuity in the current paths of a composite electrode. This contrasts with the nanomorphology of hard carbon, which is less ordered than graphite but composed of interconnected regions of stacked carbon nanosheets [17]. Smaller isotropic carbon black particles can provide short-range electrical bridges between large electrochemically active particles, provided the former are finely distributed and physically connected with active material surfaces.

Electrically conductive carbon blacks are composed of nanoscale graphitic planes, which are concentrically arranged into layered spheres with typical diameters ranging from 25 nm to 100 nm [18]. The effectiveness of such carbon blacks as conductive additives is associated with porosity and/or with graphitic edge-plane availability across the outer surface [19]. These nanoscale features also yield a high surface area (Brunauer–Emmett–Teller specific surface area (BET SSA) in the range of 45 to 72 $m^2$ $g^{-1}$ [20]), favoring aggregation into larger particles. The occurrence of larger (>1 μm) agglomerates can be disrupted during the slurry mixing process to produce electrodes with finely branched electronic networks [21].

Integrating carbon black into electrodes requires mixing of slurry components using sufficient agitation energy to blend thoroughly while avoiding damage to larger active material particles and facilitating close associations between different particle types [22,23]. Besides agitation speed, shear forces during slurry mixing are affected by multiple variables, including viscosity and particle characteristics [24,25]. Differences in rheology, mixing devices and particle dimensions for each electrode production procedure, and the difficulty of estimating shear forces within a slurry during agitation, prevent identifying an ideal range of mixing energy, even for a typical electrode formula. Ineffective electrode slurry mixing can result in a coarse conductive network, either favoring a few long-range paths that bypass regions of active material or inadequately connected short-range paths between electrically isolated regions (Figure 1).

Optimizing the amount of conductive additive added into an anode formula therefore depends not only on the particle characteristics of the additive but also on slurry mixing parameters, binder properties and particle characteristics of the active material. The multivariate nature of these physicochemical interactions has been well-described for typical electrodes used in Li-ion cells [26,27]. Comparably fewer studies have been published about the requirements for conductive additives in Na-ion electrodes [28].

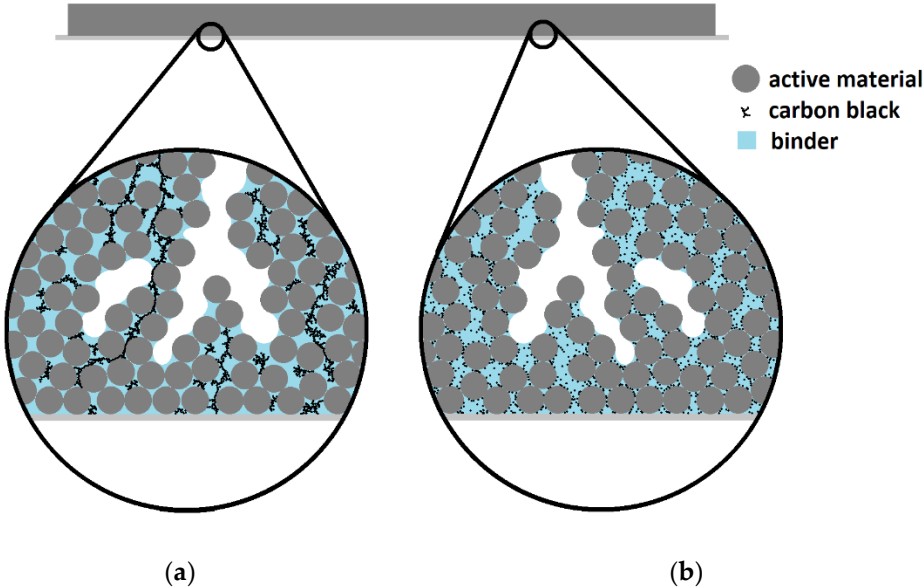

**Figure 1.** Conceptual cross-sectional diagram of a hard carbon anode film illustrating the inefficiencies in: (**a**) short-range and (**b**) long-range conductive additive networks.

Pfeifer et al. [29] evaluated the performance of six different conductive carbons in antimony-based anodes for Na-ion cells, comparing half-cell performance to precursor particle characteristics and concluding that the smallest and roundest carbon particles were best-suited as additives. The importance of nanocarbon morphology may be useful for evaluating additives for electrodes made with other active materials, though the electronic characteristics and resultant embedded current-carrying network of each candidate material was not investigated [29]. An emphasis on morphology was also made by Babu and Pyo [30], who proposed carbon nanotubes as a microstructure-enhancing co-active material in hard carbon anodes by matching pore distribution characteristics to half-cell capacity. The decreased impedance in the carbon nanotube-containing electrodes was attributed to the improved charge transfer in the pores, though another possible explanation was that the material acted as a conductive co-additive, as demonstrated in Li-ion electrodes [31]. Similarly, Zhong et al. (2022) [32] applied graphite nanoribbons as a conductive structural and sodium intercalation enhancer in nano-spherical hard carbon anodes, but this was in addition to the conductive carbon black already accounting for 10% of the electrode's dry composition. This contrasted with commercial hard carbon electrode formulas, which apply much lower proportions of conductive additives since this component displaces hard carbon in the electrode.

Ledwoch et al. [33] partially substituted carbon black with zeolite to achieve higher cycling stability and rate performance while maintaining microporosity and low binder content. Electrochemical characterization after cycling confirmed that the proposed formula improved ionic conductivity through the solid electrode interface (SEI), though these methods could not deconvolute the additive's effects on electronic conductivity through electrodes or at the electrode–current collector interface [33]. A four-point-probe measurement was used to assess the change in the sheet resistance of electrodes with zeolite content, but this exclusively quantified the current resistance along the paths parallel to the current collector, which are perpendicular to the current paths in a cell.

A plethora of variables complicate the exclusive use of electrochemical methods to evaluate the effectiveness of conductive additives in electrodes. The known electrochemical activity of carbon black contributes to both measured faradaic output [34] and to side reactions such as the formation of an SEI. The SEI's composition and stability are dependent on a long list of variables from electrolyte materials to initial cycling parameters [35], which are not standardized in electrode testing, preventing comparisons between published studies. Although cycling experiments can be useful in assessing the effect of novel

materials on cell capacity and initial coulombic efficiency, they are less useful for monitoring internal resistance [12], and the post-cycling impedance associated with an SEI formed by galvanostatic methods masks that from non-interfacial sources [36].

While direct ohmmeter tests of dry electrodes can avoid the electrochemical variables that obscure the quantification of bulk electronic resistance, a conventional four-point probe test quantifies only the most conductive paths along the length of an electrode film, which is unlikely to be representative of resistive gradients to currents passing through the depth of an electrode. Edge effects can also be expected to affect the accuracy of a four-point measurement, which assumes effectively infinite plane dimensions despite current density being non-uniform across the area of an electrode sheet. To overcome these limitations, Westphal et al. developed a custom test stand with controllable pressure to quantify through-resistance while standardizing contact pressure [37]. The measured resistance represented more closely the current path through the electrodes, but such specialized equipment is unlikely to become widely available or simple to construct.

To resolve these obstacles to the electronic characterization of hard carbon electrodes, a practical alternative resistance measurement for the evaluation of conductive additives is proposed in this work. By adhering pin contacts above and below electrode slabs poured onto current collector foil, a two-point ohmmeter measurement was used to quantify the electronic resistance to current through battery-grade hard carbon anodes. The method was applied to thick samples of electrodes produced with hard carbons from two different sources, which were impregnated with increasing amounts of conductive carbon black, to quickly determine the long-range electronic conductivity. This novel method can be used to assess the long-range electronic conductivity of any slurry with similar flow properties, providing a practical and low-cost characterization tool for cell researchers formulating electrodes with a range of chemical compositions.

The results from the two-probe ohmmeter tests were also compared to impedance levels measured on symmetrical coin cells, which were made from matching hard carbon electrodes, to elucidate the corresponding short-range current paths. Uncycled electrodes were tested to minimize the sources of ionic resistance during the quantification of electronic resistance in a relevant electrochemical system. By matching the dry resistance results with the electrode interfacial impedance spectra, both long-range and short-range electrical connections were used to assess the two contrasting slurry mixing protocols. Our findings strongly recommend that traditional characterization methods, such as galvanostatic half-cell cycling and the chemical spectroscopy of pure active material, cannot describe some effects of the critical formulaic and process variables, such as conductive additive and mixing speed, which can be more reliably examined by applying our practical techniques.

## 2. Materials and Methods

### 2.1. Materials

All slurries started from a solution of sodium carboxymethylcellulose, or CMC (Sigma Aldrich, Merck, St. Louis, MO, USA), degree of substitution 0.9, molecular weight 250,000 Da), in de-ionized water (Millipore, Burlington, MA, USA), which was dissolved with a magnetic stirrer and rested overnight. The conductive additive, Super C65 carbon black (Imerys S. A., Paris, France), was added into the solution and mixed with either a single-head propeller-type stirrer (Stuart SS10, Cole Parmer Instrument Company Ltd, St. Neots, UK) with a 50 mm diameter blade spun at 1500 rpm for 10 min or a Speedmixer (DAC 150.1 FVZ-K, Synergy Devices Ltd., High Wycombe, UK) at 3500 rpm for 10 min. Subsequently, active material composed of either hard carbon from Kuraray (Kuranode Type 2, 9 μm, Kuraray Co. Ltd., Kita-Ku, Osaka, Japan), hard carbon from MTI (SIB-BHC300, MTI Corporation, Richmond, CA, USA) or graphite (Aldrich 907154) was added and mixed for five minutes according to each of the two methods described (Table 1). After resting overnight, the slurries were re-mixed for 1 min at 1000 rpm with the propellor-type stirrer before aqueous styrene-butadiene rubber, or SBR, suspensions (EQ-Lib-SBR, MTI Corporation, Richmond, CA, USA) were added and mixed into the slurries with the propellor-type

stirrer at 1500 rpm for 5 min, immediately followed by tape-spreading electrode films. The solid fractions of the slurries were maintained between 55% and 60%, composed of 2% each of CMC solid and SBR solids and carbon black adjusted in a range of proportions from 0% to 5%.

**Table 1.** Particle surface area and size range of hard carbon and carbon black applied herein.

| Material | Surface Area (m$^2$/g) | Mean Particle Diameter | Reference |
|---|---|---|---|
| Hard carbon, Kuranode | 4 | 9 μm | [38] |
| Hard carbon, MTI | $\leq$8 | 3–7 μm | [39] |
| Carbon black, Super C65 | 62 | 150 nm | [20] |

The surface characteristics of both these commercial carbon products, such as Raman spectra, are well established in other publications ([40,41], respectively, for the spectra of Kuranode and C65 carbon black). Spectroscopic information available in the literature confirms both hard carbon and carbon black are composed of disordered graphitic sheets at the nanoscale.

*2.2. Electrochemical Sample Testing*

Each slurry sample was spread into a film by tape casting onto aluminum foil (15 μm, MTI Corp.) using a glass bar (7 mm diameter) over a depth of one or two tape layers (Scotch® Magic™, Greener Choice). The films were dried between 24 and 96 h without added heat or airflow. The thicknesses of films were determined with a micrometer (0–25 mm and 0.001 mm, Mitutoyo Ltd, Andover, UK). The films verified to be between 35 μm and 65 μm thick were subsequently baked at 120 °C in a vacuum for a minimum of 8 h before being cut into 16 mm discs in an argon environment. The cut electrodes and bare foil discs were weighed to determine the active material loading, which varied according to the thickness and displacement of the hard carbon with the added carbon black. The active material loading of individual electrodes ranged from 3 mg cm$^{-2}$ to 6 mg cm$^{-2}$ (the values are shown in Appendix B).

Symmetrical coin cells were used to apply two identical electrodes, each backed by a 0.5 mm steel spacer, while the half-cells contained one hard carbon electrode backed by one 0.5 mm steel spacer and a sodium metal disc (Sigma Aldrich, Merck, Montcuq, France) cut to a diameter of 15 μm (using the sodium rolling method shown in [42]). All CR2032 cells applied one steel wave spring, one layer of polypropylene separator (Celguard 2500) and one layer of glass fiber (Ohaus Europe GmbH, Nänikon, Switzerland) cut to a 16 mm diameter. The hard carbon electrodes and separators were pre-soaked for a minimum of 4 h in electrolyte to avoid the impedances caused by inadequate surface wetting, and the resulting volume-loading exceeded 100 μL for all cells based on the measured retention of fluid in a glass-fiber disc. The electrolyte used in all cells consisted of 1 M NaClO$_4$ mixed into a blend of either ethylene carbonate (EC) and propylene carbonate (PC) or EC and dimethyl carbonate (DMC) at a 50:50 *w/w* ratio (using the electrolyte mixing method shown in [43]). The electrolyte density was found to be 1.84 g ml$^{-1}$ at average glovebox pressure and temperature conditions. The half-cells were galvanostatically cycled with a Maccor model 4300 battery tester at 300 μA (to achieve at minimum 30 mA g$^{-1}$). The cells made with Kuranode hard carbon electrodes containing 2% carbon black were sodiated and desodiated 5 times between 2.0 V and 0.005 V vs. a sodium metal cathode, followed by a final sodiation to 0.005 V. The galvanostatic cycling results are included (Figures A7 and A8).

Electrochemical impedance spectroscopy (EIS) was performed using an Ivium CompactStat by applying waveforms from 100 kHz to 0.1 Hz at a 0.01 V amplitude biased at an open circuit potential (obtained over 15 s before each measurement). The measurement values recorded for real impedance (Z′), imaginary impedance (Z″) and impedance modulus (|Z|) were analyzed without fitting to an equivalent circuit, with the first recorded Z′ value (at an applied frequency of 100,000 Hz) used as a proxy for the series resistance (Figure 2a). The maximum modulus values (recorded at an applied frequency of 0.1 Hz)

were averaged, and standard deviations were calculated for each sample type (using Excel 365). A minimum of two identical cells of each type were tested across a minimum of two different days to ensure the repeatability of the results. The effects of variables such as temperature, humidity, electrolyte aging, wetting time and human technique were mitigated by constructing and testing different types of cells on the same day.

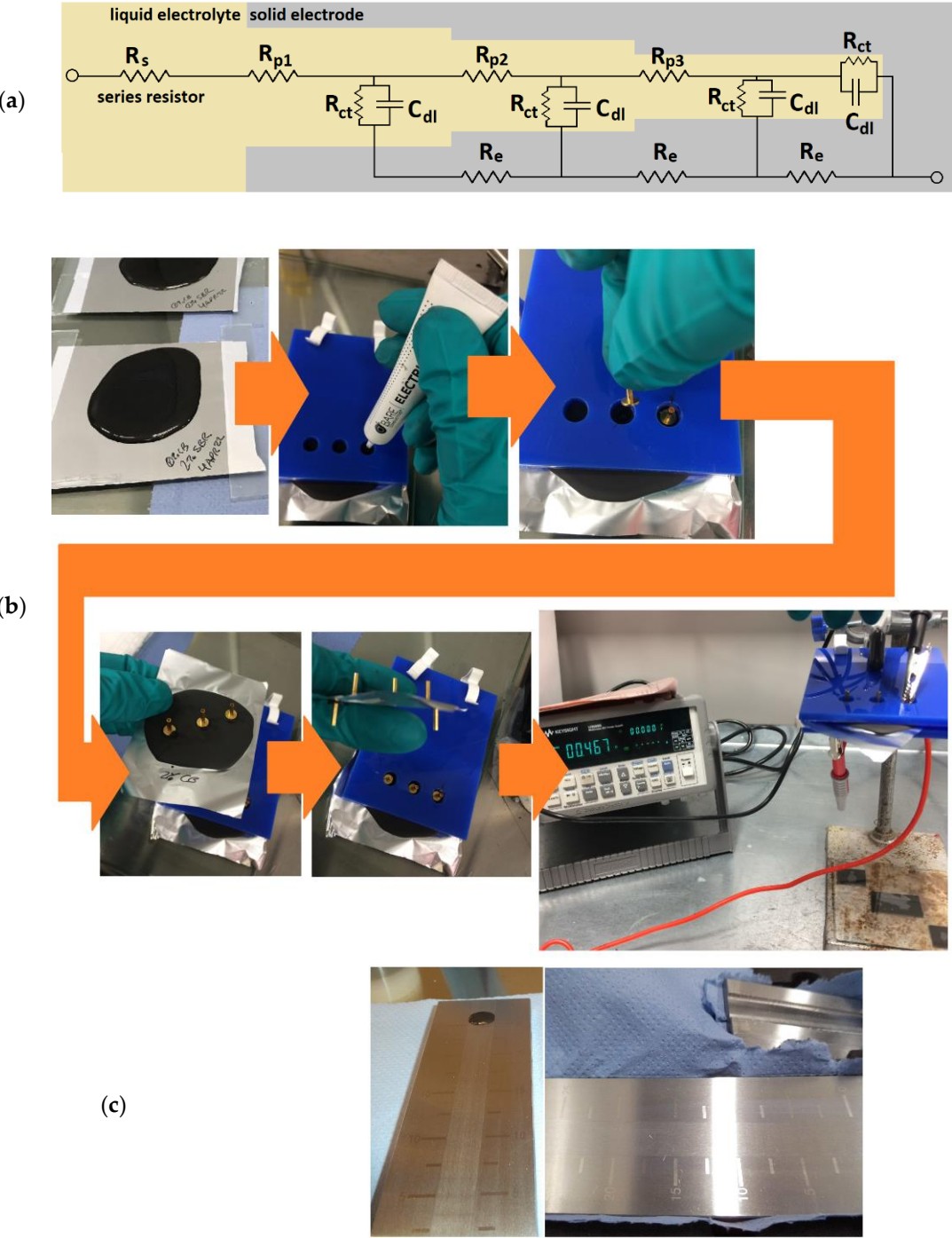

**Figure 2.** Methods applied herein: (**a**) equivalent circuit of un-sodiated porous hard carbon electrodes; (**b**) slurry slab preparation showing the custom stencil; and (**c**) Hegman gauge and blade before sweeping a sample. $R_{p1-3}$ represents the pore resistances, $R_e$ represents the resistance in a solid electrode, $R_{ct}$ represents the charge transfer resistance and $C_{dl}$ represents the double-layer capacitance.

### 2.3. Alternative Slurry Testing

Sample slabs of between 1 and 2 mm thick were produced by pouring approximately 25–50 mL of each sample slurry onto aluminum foil and drying for a minimum of 48 h. The slabs were further prepared by applying conductive paste (Electric Paint, Bare Conductive, London, UK) to attach gold-coated contact pins (Coda Systems Ltd., IPP2-6, Essex, UK) to three points on each upper surface and two points on each current collector. The distances between the points were standardized between samples by using a custom vinyl stencil (Figure 2b).

The resistance through the slabs was measured with a multimeter (U3606B, Keysight Technologies, Santa Rosa, CA, USA) used as a two-point ohmmeter. The two-point resistance measured through the electrode slabs was recorded between six possible combinations of upper and lower contacts. For each slab, the average values were normalized to a sample thickness of 1 mm. A minimum of two separate slabs made from slurries mixed on different days were produced for each percent of carbon black in the Kuranode hard carbon slabs. The *p*-values were determined with a *t*-test (Excel 365, 2-tailed, 2-sample unequal variance) and compared to 0.05 for statistical significance.

Slurry particle tests were performed with a Hegman gauge (Baoshishan grindometer 0–25 μm). After all slurries were rested for a minimum of 48 h, each was applied to the upper end of the gauge before sweeping with a steel blade angled at approximately 60 degrees. The gauge was photographed within 10 s and rinsed with deionized water and ethanol between uses. Photographs were used to assess the suspended particle dispersion by recording the beginning and end of visible color fade (Figure 2c).

The hard carbon electrode samples produced as described were also characterized with scanning electron microscopy using a Hitachi (Tokyo, Japan) tabletop scanning electron microscopes (SEM, TM4000) at 2000× magnification and 5 kV, as well as a Keyence (Osaka, Japan) VHX-7000 optical microscope at 2500× magnification.

## 3. Results and Discussion

### 3.1. Conductive Additive

The impedances of the hard carbon electrodes with increasing quantities of conductive additive were assessed initially since differences in the electrical conductivity levels proportional to the carbon black content were expected to be detectable when the changes in ionic conductivity were minimized (raw data in Supplementary Materials). In this configuration, the impedance spectra of the symmetrical (non-sodiated) coin cells described current paths through the two matching hard carbon/electrolyte interfaces, which could be modelled by a transmission-line equivalent circuit (Figure 2a). The tortuous microstructure of the solid–liquid interface caused differences in the RC values along the depths of irregularly shaped pores, preventing the accurate fitting of the components in the model. In this case, only the series resistor could be estimated with relative confidence since this value can be extracted from the first real impedance in the spectra, provided the upper frequency range applied is relevant to the system and unaffected by artifacts and noise.

A trend was observed at the low-frequency end of the spectra, where electrodes with more conductive additive demonstrated incrementally lower real and imaginary impedance levels (inset to Figure 3a). Comparing the impedance modulus values further illustrated this link, showing an inverse relation between the percent of carbon black and the maximum |Z| at each frequency (Figure 3a,b). This might be explained by more than one phenomenon related to carbon black, including microstructural changes to pore shapes [44], increased electrochemical activity in carbon black surfaces [45] and/or increasing active material surface area through binder redistribution [46]. While each of these variables could affect the performance of anodes in a secondary cell, they obscure the quantification of electronic conductivity as the primary purpose of a conductive additive and of this work.

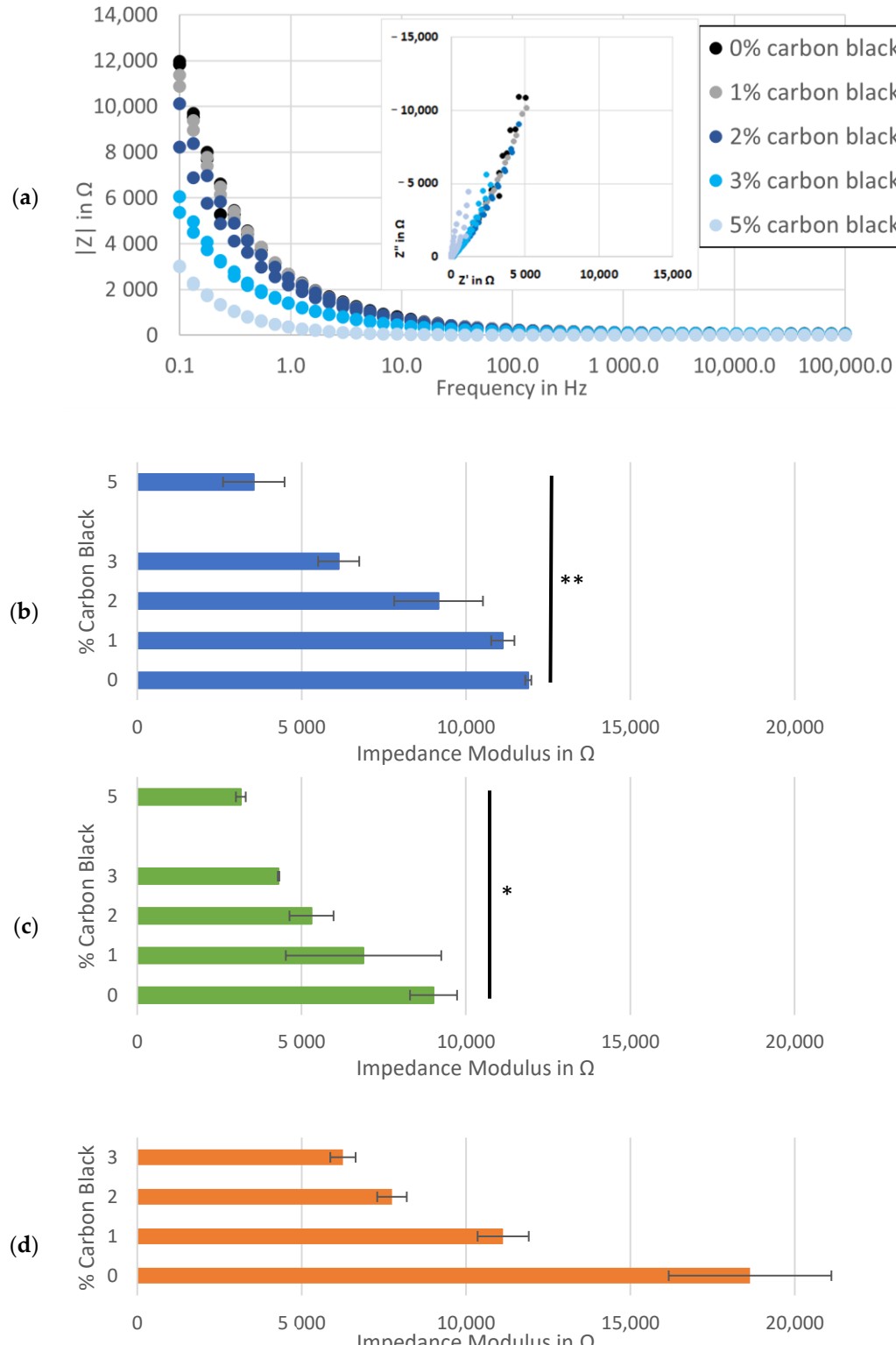

**Figure 3.** Impedance analysis of symmetrical cells with matching Kuranode hard carbon electrodes, showing impedance magnitude: (**a**) as a Bode plot with Nyquist on the inset; (**b**) average at 0.1 Hz in the standard electrolyte; (**c**) average at 0.1 Hz with substituted low-viscosity solvent and (**d**) average at 0.1 Hz of cells using MTI hard carbon electrodes. the error bars indicate the standard deviations, and * $p < 0.05$ and ** $p < 0.005$.

The high-frequency values in the impedance spectra were expected to relate more closely to the resistance through the solid phase of the electrodes since this value was

included in the sum of the resistances modelled by the series resistor. By standardizing other contributors to series resistance, such as separator thickness, electrolyte composition, cable connectors and stack pressure, increases in conductive additive were hypothesized to yield decreases in resistance to current through the electrodes, but this was not observable as a change to the series resistance (Figure A1). This real resistance varied between the cells containing the same proportion of carbon black while the average series resistance was similar, regardless of the carbon black content.

The independence of the series resistance to the changes in conductive additive was confirmed by repeating the impedance tests on the cells made with an alternative electrolyte composition. By substituting the low-viscosity solvent, PC, with an alternative also used in Na-ion cells, DMC, the impact of the solvent properties on the system resistance could be illustrated, even in the absence of an SEI layer (Figure 3c). When DMC was used, series resistance decreased and the gradient of imaginary to real impedance was steeper (Figures A2 and A3).

The much lower viscosity of DMC compared to PC [47] would be expected to increase diffusion efficiency through tortuous channels, which explained the decreased real impedance at both the low frequencies (through electrode pores) and the high frequencies (through separator pores). Although the possibility of electroactive activity between DMC and hard carbon exists, this was assumed to be less likely to cause the observed impedance spectra than the differences in the fluid mechanics between the two evaluated electrolyte compositions. Confirming this explanation would require a more detailed characterization of the pre-SEI formation interfacial reactions when applying DMC or PC, which was considered to be beyond the scope of this work. However, these results confirmed the complexity of evaluating the direct effects of carbon black on the electronic conductivity of hard carbon electrodes while in the presence of electrochemical variables.

This complexity motivated the development of a "dry" test to quantify the resistance to currents through hard carbon electrodes, which could also include the resistance of the solid-to-solid electrode/current collector interface. Traditional along-plane tests such as four-point resistance are not representative of through-plane current paths through the thickness of an electrode and the interface with current collector while resistances measured through thin film electrodes are generally too small to be measured directly by an ohmmeter. These problems were addressed by pouring thick 'slabs' of electrode slurry onto current collector foils, followed by attaching highly conductive contacts onto both the upper electrode surfaces and underneath the current collector foil to simulate solid-to-solid contacts within a cell (Figure 2b).

Resistances through sample slabs, including interfaces with the contact and current collector, were normalized to a 1 mm thickness and compared between three active materials. The interfacial resistances between gold-coated contacts, conductive ink and sample surfaces were assumed to be effectively equal for each sample and deemed negligible for the purpose of comparing the samples prepared identically. The percolation of conductive additive demonstrated a sharp decrease in resistance of between 0% and 1% carbon black while similar trends were observed for the hard carbons and graphite (Figure 4). Formulations containing 1% to 5% of carbon black maintained resistances of less than $0.5 \ \Omega \ \mathrm{mm^{-1}}$ for all three active materials. Although the resistance of the dry electrode material could be approximately halved by adding 5% carbon black to either hard carbon, the formulas containing no conductive additive at all remained well below $1 \ \Omega \ \mathrm{mm^{-1}}$. This contrasted with the graphite slab, which appeared to be less conductive to electronic currents than hard carbon when no carbon black was present.

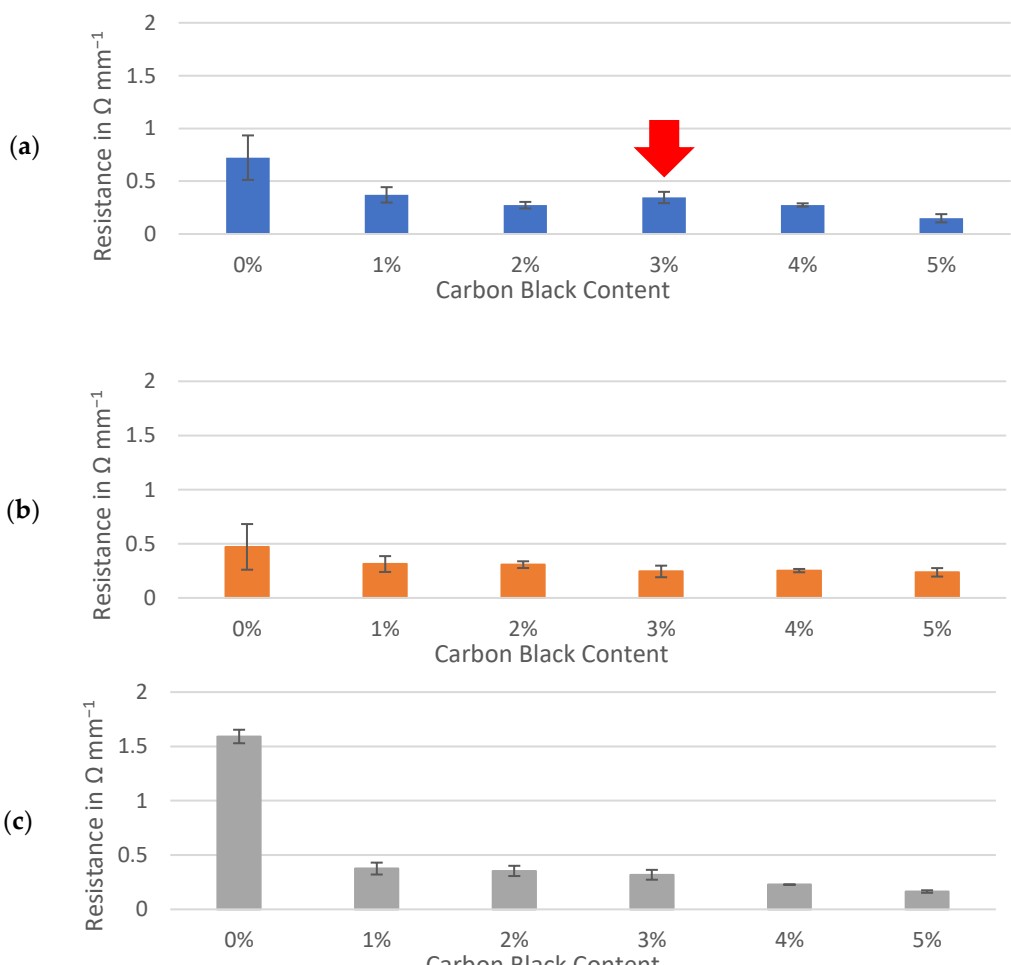

**Figure 4.** Through-resistance of: (**a**) Kuranode; (**b**) MTI hard carbon and (**c**) graphite electrode slabs on current collector foil. The error bars indicate the standard deviations.

With the Kuranode hard carbon, a significant and unexpected increase in resistance occurred when the conductive additive was increased from 2% to 3% (Figure 4a, red arrow). Upon further increasing the carbon black from 3% to 4%, the through-resistance returned to values not significantly different from those at the 2% conductive additive. This fluctuation in the percolation curve, which related conductive additive to long-range resistance, suggested negligible advantages or even disadvantages with carbon black increases within this range. The impedance modulus, which represents both short-range and long-range electronic network connections in a thin film, did not fluctuate when carbon black was added at these proportions (Figure 3b–d). Since the dry slab sample tests could only quantify the long-range electronic paths through the electrodes, the fluctuations could have been an artifact created by differences in the through-resistance at different regions along the electrode plane, revealing a possible limitation of applying thick slabs as electrode samples for the proposed method.

This percolation curve fluctuation was not observed with the MTI hard carbon, nor was it observed with the graphite, presenting the possibility that the anomaly was specific to the particle interactions between the Kuranode hard carbon and the conductive carbon black. The differences in the primary particle sizes between the two hard carbons could account for the differences in the long-range electronic resistances, as described by Indrikova et al. [48]. In the case of the Kuranode, increasing the measured resistance at 3% carbon black could indicate ineffectiveness of the conductive additive at this proportion, or the electronic network may not have been distributed efficiently. The latter possibility was consistent with the continuation of the lower resistances when carbon black was further added to

5% of the dry material since this extra content could have provided bridges between the otherwise disconnected short-range networks. Large agglomerations of carbon black could decrease the efficiency of the long-range connections, and a suspected cause of this problem was ineffectively mixing the slurry.

*3.2. Mixing*

To evaluate the effectiveness of the selected slurry mixing protocol while contributing to the validation of the dry slab resistance method, these tests were repeated on both types of hard carbon after blending the slurry at a higher speed (3500 rpm vs. 1500 rpm). The higher speed mixing was achieved using a centrifugal-type mixer, which is also assumed to distribute shear forces more evenly across a slurry container than a propellor-type stirrer. Increasing the speed and energy distribution of the mixing at the conductive additive and hard carbon stages was expected to ensure the disruption of any unbroken agglomerations of carbon black that might have remained in the original electrode samples, though this practice also risked unanticipated effects that could result from higher shear forces.

At a mixing speed of 3500 rpm (Figure 5a,b), the impedance spectra of the hard carbon continued to demonstrate a trend of decreasing moduli with increasing quantities of carbon black (as seen in Figure 4a,b). Unexpectedly, the average impedance moduli at each percent of carbon black in the electrodes mixed at 3500 rpm often exceeded that of those mixed at 1500 rpm, though this difference was only statistically significant for the MTI hard carbon ($p$ = 0.0075 at 1%, 0.0476 at 2% and 0.0237 at 3% carbon black). At both speeds, the higher impedance moduli for the MTI hard carbon could be attributed to this material's smaller particle size compared to that of the Kuranode hard carbon, which can pack more densely into the same electrode volume (Table 1). When the mixing speed was increased, any pulverization of the active material particles caused by the higher shear forces would have enhanced this densification, an effect that has been observed for graphite anodes [49,50].

Comparing the through-resistances of the dry sample slabs also revealed changes caused by the mixing speed (Figure 5c,d). For both hard carbon types, when higher shear forces were applied in mixing, the long-range resistance of slabs with 0% carbon black decreased to below 0.5 $\Omega$ mm$^{-1}$. For both hard carbons, in most cases, the average through-resistance after mixing at 3500 rpm was lower than that mixed at 1500 rpm, but this was not consistent. While all higher-shear-mixed samples yielded through-resistances of between 0.23 and 0.47 $\Omega$ mm$^{-1}$, the Kuranode formula applying 2% carbon black was the only through-resistance not significantly different between the mixing speeds ($p$ = 0.0754).

The anomalous percolation fluctuation previously observed at 3% carbon black for the Kuranode did not occur at the higher mixing speed. Although resistance through the Kuranode slabs appeared to increase with 1% to 2% carbon black, this was not statistically significant ($p$ = 0.600). For the MTI hard carbon, no decreases in resistance with added carbon black were apparent when higher speed mixing was applied, and the through-resistance remained statistically constant until 3% carbon black was added. Using this mixing protocol with MTI hard carbon, no advantage was gained in the long-range resistance by adding 2% or less conductive additive. The greatest impact to long-range resistance caused by increased mixing speed was observed in the samples with no carbon black, suggesting that the mechanism of this change was unrelated to the electronically conductive additive network. This supported the possibility that particles of active materials were pulverized at the higher mixing speed, resulting in the occurrence of smaller particles of hard carbon which would increase the packing density in the electrode. To investigate this possibility, microscope imagery of hard carbon electrode surfaces absent of carbon black was obtained (Figure 6). While no apparent differences between mixing speeds could be observed in the Kuranode samples (Figure 6a), the MTI hard carbon samples appeared to be more tightly packed when examined using optical microscopy (Figure 6b). This densification of the MTI-based electrodes mixed at higher speeds was clear in the images taken with electron microscopy (Figure 6c). Tighter packing caused narrower pores, leading to the increased impedance moduli observed while also increasing the connections

between the conductive particles to produce lower long-range resistances. Fragments of the active material particles that were pulverized by higher mixing shear forces could have acted as electronic bridges, effectively replacing carbon black in the conductive network.

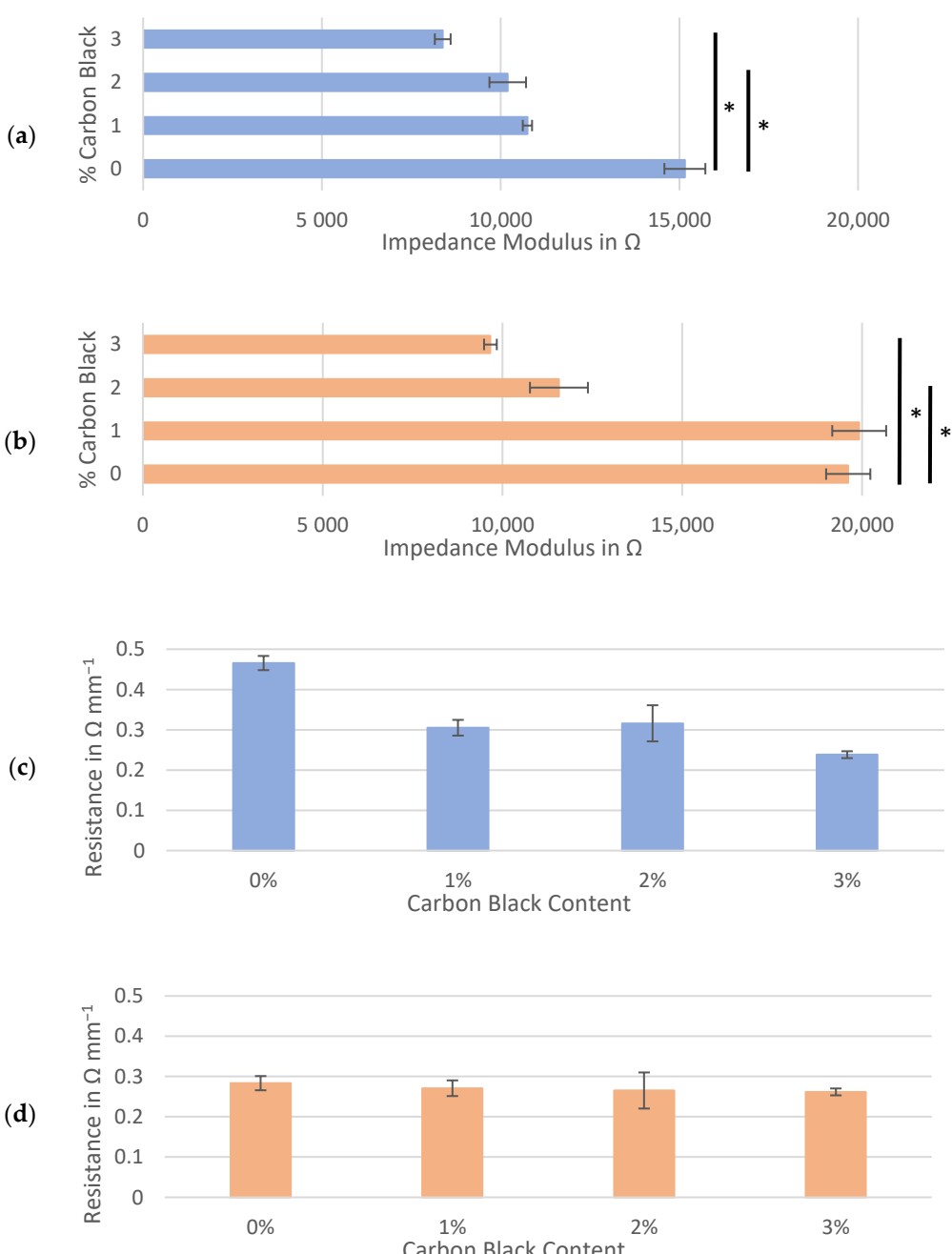

**Figure 5.** Hard carbon slurries mixed at 3500 rpm: average impedance modulus at 100 kHz for the symmetrical cells composed of (**a**) Kuranode and (**b**) MTI hard carbon, and the corresponding through-resistance of the (**c**) Kuranode and (**d**) MTI hard carbon. The error bars indicate the standard deviations, and * $p < 0.05$.

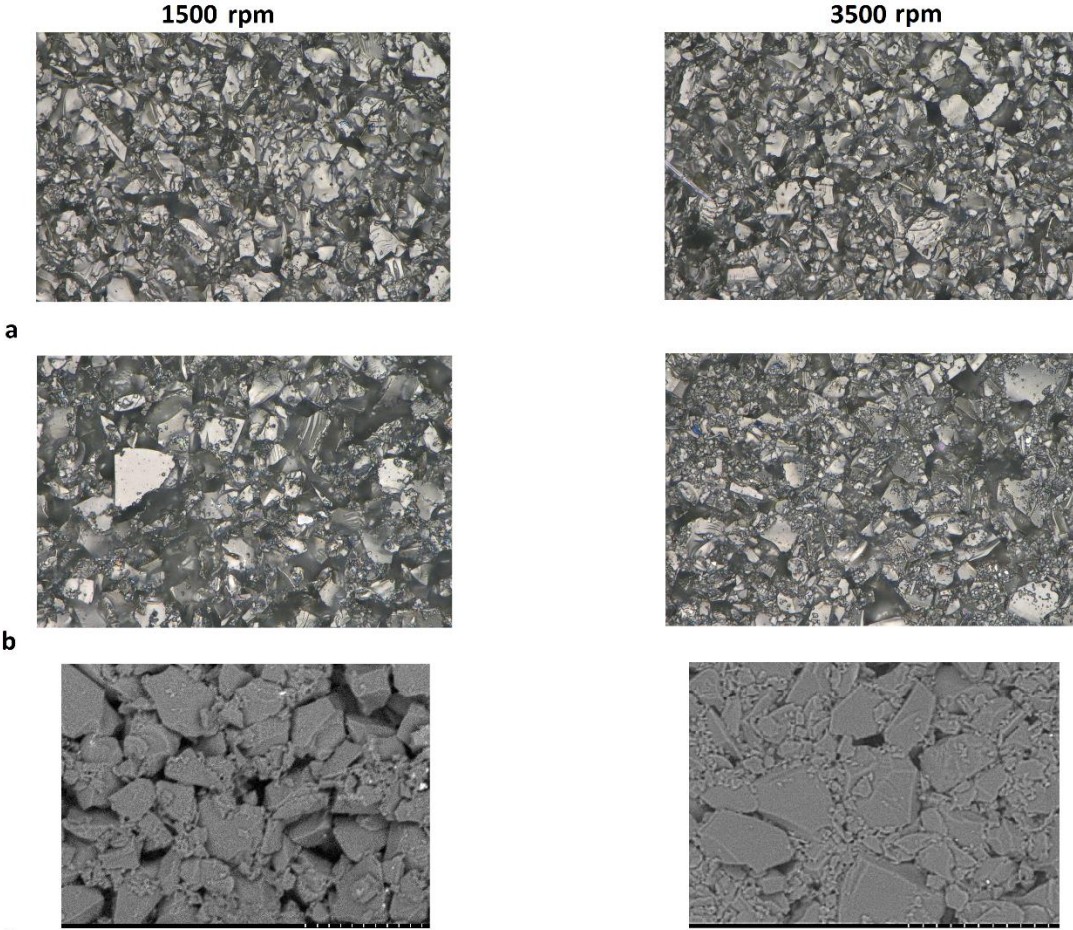

**Figure 6.** Images of the electrodes absent of carbon black mixed at 1500 rpm and 3500 rpm: (**a**) Kuranode hard carbon optically magnified 2500×, and (**b**) MTI hard carbon. SEM images, magnified 2000×, of the (**c**) MTI hard carbon samples. The scale bar represents 20 μm.

Differences in the electronic resistance between the two types of hard carbon could also be enhanced by contrasting the surface areas between particles of these two active materials since this can affect associations with other particles and with binders. For example, active material particles can form agglomerates insensitive to lower-energy mixing [51], which would not be apparent in microscope imagery. For graphite anode slurries without conductive additives, the sequence of mixing was found to affect the adsorption of CMC and SBR to the active material surfaces [52].

To further explore whether the binder associations differed between the two hard carbons, the slurries used in the production of the films were applied to a Hegman gauge. When no carbon black was present, a visible difference in the effective dispersion could be observed between the MTI slurry mixed at 1500 rpm and that mixed at 3500 rpm (Figure 7a). The effect of mixing speed was much less apparent in the slurries containing 3% carbon black, which corresponded to the results from the dry resistance tests of the MTI hard carbon ($p = 0.00002666$ at 0% carbon black and $p = 0.0757$ at 3% carbon black). In the Kuranode slurries, mixing speed did not appear to affect the dispersion assessed with this technique (Figure 7b), despite the dry slab tests indicating significantly higher long-range resistances using 3500 rpm compared to 1500 rpm ($p = 0.0106$ at 0% carbon black and $p = 0.00000129$ at 3% carbon black).

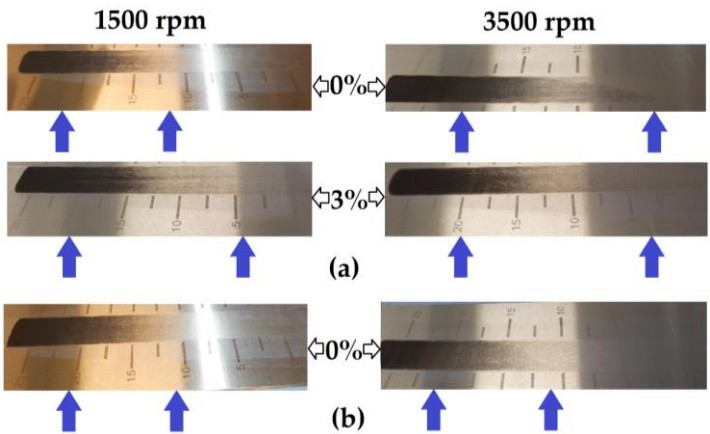

**Figure 7.** Hegman gauge tests of the slurries: (**a**) MTI hard carbon mixed at 1500 rpm and 3500 rpm; (**b**) Kuranode mixed at 1500 rpm and 3500 rpm.

Although the Hegman gauge results cannot be accurately quantified (see blue arrows), this alternative method confirmed that higher shear mixing changed the properties of the MTI slurry, independent of the electrical and drying variables. The dispersion with this technique also supported the electrochemical data, suggesting that the Kuranode slurries were less affected by the increases in the shear forces of mixing in the tested ranges since the impedance moduli compared between the mixing protocols differed significantly for the MTI electrodes but not the Kuranode films. The optimal slurry mixing speed and device, therefore, cannot be assumed to be common across all types of hard carbons.

The addition of conductive carbon black was found to decrease both the long- and short-range resistances of the electrodes from both hard carbons, regardless of the mixing technique. Both types of hard carbon electrodes were electronically conductive below 1% added carbon black, and they were more electronically conductive than the graphite electrodes produced identically. These outcomes suggest any hard carbon anode formulation should be optimized for each material source, particularly in relation to the particle sizes of the active materials. The ideal proportion of conductive additive in hard carbon electrodes should also be based on the rate requirements of the design application since advantages in electronic resistance might be outweighed by the influence of carbon black on pore microstructure, binder-particle coverage and slurry rheology.

## 4. Conclusions

This work validated a novel approach to assess the effectiveness of blending carbon black into hard carbon anodes as a conductive additive. Combined with symmetrical cell impedance moduli as markers of the charge transfer resistance within the pores, the utility of adding C65 carbon black into electrode formulations could be directly linked to decreases in both long-range and short-range current paths between the electrochemical interface and the current collector. The suggested techniques avoided the disadvantages of conventions such as voltammetry and cell cycling, which form a resistive SEI layer dependent on multiple parameters and prevent comparisons between independent studies caused by several variables, such as electrolyte solvent.

These results suggest that slurry mixing protocol can be more critical to manipulating electronic resistance than the amount of conductive additive in a hard carbon anode formula. In the electrode samples made from both hard carbons with no conductive additive, the long-range resistances decreased significantly when high shear force mixing was applied. In contrast, the low-frequency impedances of the samples made from slurries mixed with high shear forces were higher than those mixed at lower speeds. This increased impedance was attributed to the pulverization of some active material particles when higher mixing energy was applied, which increased the packing density of the electrode film and consequent tortuosity of the electrode-containing pores. The application of a Hegman gauge further

confirmed the differences in the particle-to-binder interactions between the hard carbons from the two sources, which contributed to the contrasting electrode properties between these active materials.

The proposed methods can provide a quick and reliable procedure to link changes in electrode formulation or processing with solid electrode conductivity while using fewer resources than traditional cell cycling. The simplicity of these tests provided advantages in both practicality and repeatability, which allowed statistical analyses between samples that could be extended to monitor consistency over time or between electrodes mixed using different procedures. Characterizing the slurry rheology and contrasting binder types could provide further insight into the inter-particle associations of anode and cathode materials frequently used in sodium-ion cell development, with multiple potential advantages for manufacturing efficiency.

**Supplementary Materials:** The following supporting information can be downloaded at: https://www.mdpi.com/article/10.3390/coatings13040689/s1, Figure S1: Raw impedance data for cells applying Kuranode electrodes mixed at 1500 rpm using a PC solvent; Figure S2: Raw impedance data for cells applying Kuranode electrodes mixed at 1500 rpm using a DMC solvent; Figure S3: Raw impedance data for cells applying MTI electrodes mixed at 1500 rpm using a PC solvent; Figure S4: Raw impedance data for cells applying Kuranode electrodes mixed at 3500 rpm using a PC solvent; and Figure S5: Raw impedance data for cells applying MTI electrodes mixed at 3500 rpm using a PC solvent.

**Author Contributions:** Conceptualization, methodology, validation, formal analysis, investigation and writing—original draft preparation, M.A.S.; writing—review and editing, supervision, project administration and funding acquisition, J.B. All authors have read and agreed to the published version of the manuscript.

**Funding:** This research was funded by the Engineering and Physical Sciences Research Council (EPSRC), ECR Fellowship NoRESt (grant number EP/S03711X/1) and SPECIFIC Innovation and Knowledge Centre (grant numbers EP/N020863/1 and EP/P030831/1).

**Institutional Review Board Statement:** Not applicable.

**Informed Consent Statement:** Not applicable.

**Data Availability Statement:** The data presented in this study are available in the Supplementary Materials.

**Acknowledgments:** The authors would like to thank Chris Griffiths at the SPECIFIC Pilot Manufacturing Research Centre for his essential assistance with obtaining the microscopy images.

**Conflicts of Interest:** The authors declare no conflict of interest.

## Appendix A

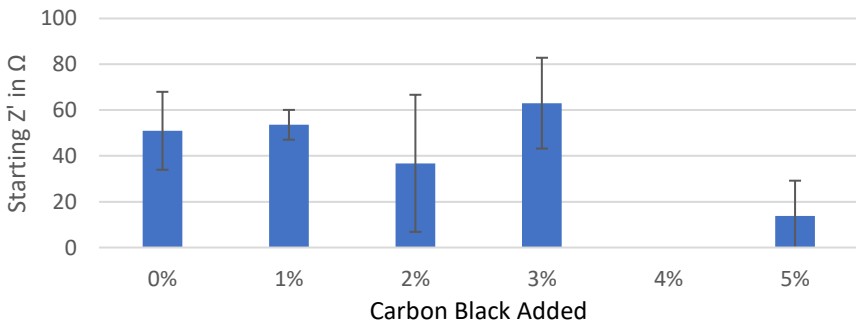

**Figure A1.** Starting real impedance of the Kuranode hard carbon electrodes according to the conductive additive content, as measured with the electrolyte containing PC as a low-viscosity solvent.

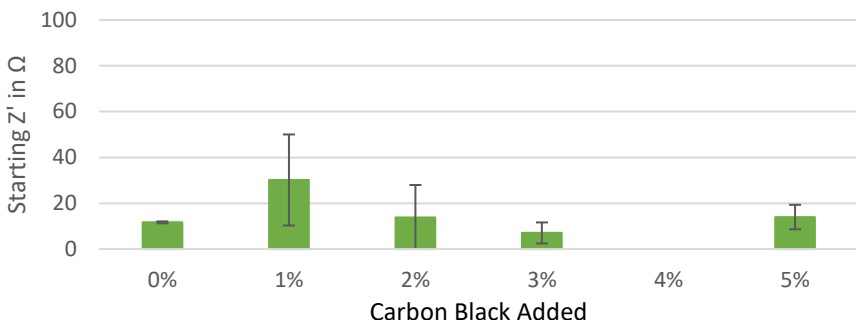

**Figure A2.** Starting real impedance of the Kuranode hard carbon electrodes according to the conductive additive content, as measured with the electrolyte containing DMC as a low-viscosity solvent.

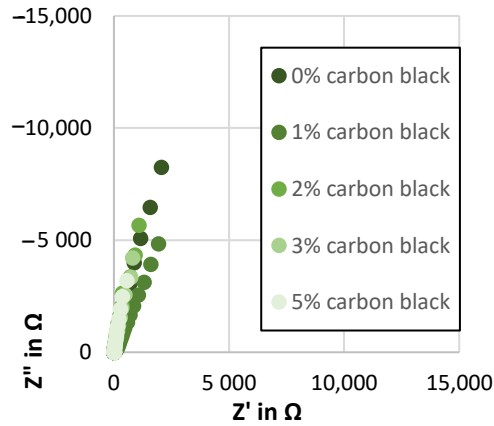

**Figure A3.** Nyquist plot showing the impedance spectra of the Kuranode hard carbon electrodes with different conductive additive contents, as measured with the electrolyte containing DMC as a low-viscosity solvent.

**Appendix B**

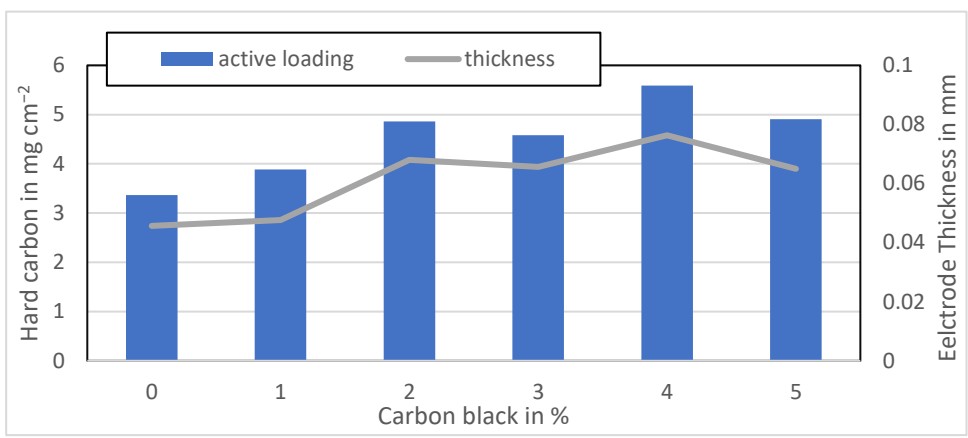

**Figure A4.** Active material loading of the Kuranode anodes mixed at 1500 rpm.

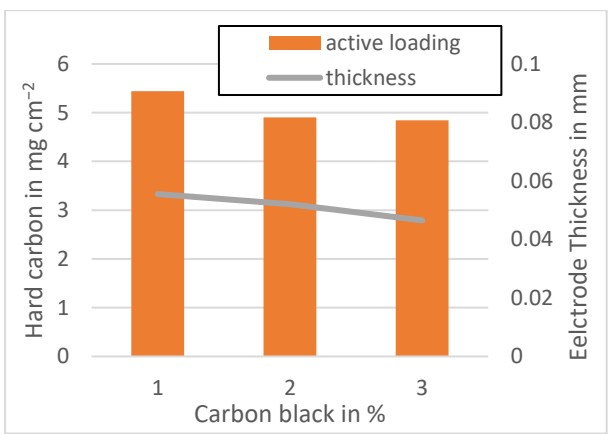

**Figure A5.** Active material loading of the MTI anodes mixed at 1500 rpm.

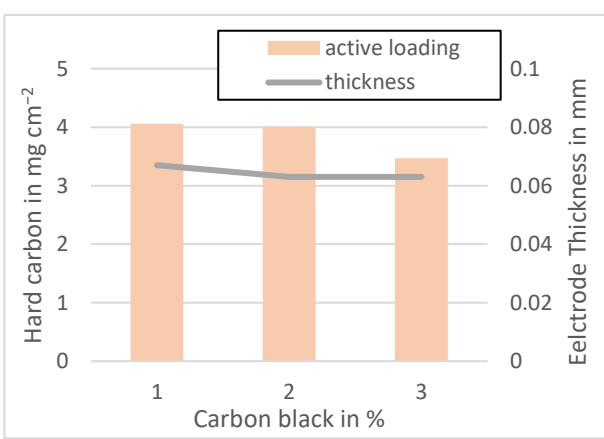

**Figure A6.** Active material loading of the MTI anodes mixed at 3500 rpm.

**Appendix C**

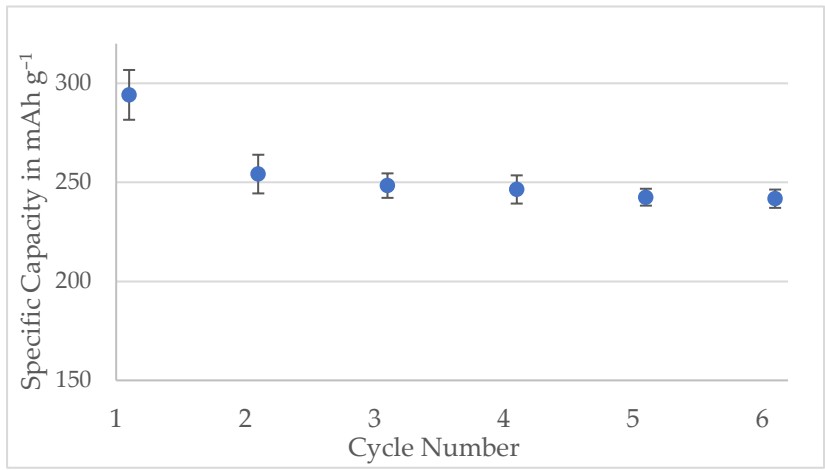

**Figure A7.** Specific capacities of the half-cells made with Kuranode hard carbon containing 2% carbon black. The bars represent the standard deviations of three cells.

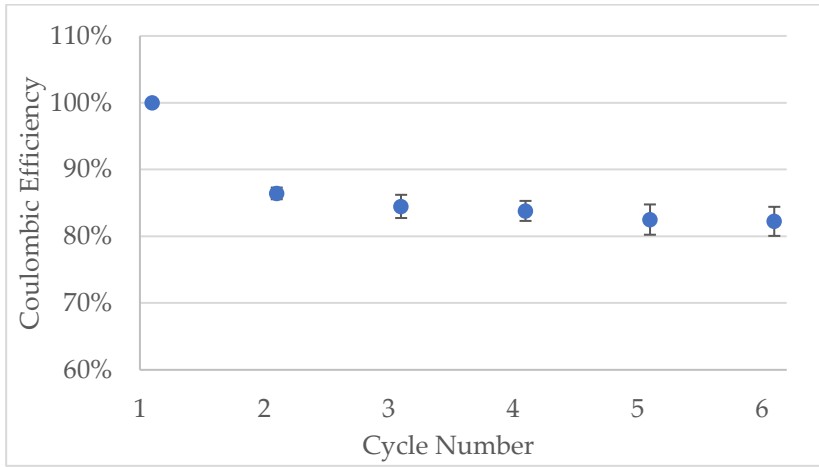

**Figure A8.** Coulombic efficiency of the half-cells made with Kuranode hard carbon containing 2% carbon black. The bars represent the standard deviations of three cells.

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
