# Peer review of "Mixing, Fast and Slow: Assessing the Efficiency of Electronically Conductive Networks in Hard Carbon Anodes"

_coatings, doi:10.3390/coatings13040689_

Round 1

Reviewer 1 Report

The manuscript is devoted to the study of the morphology and conductivity of hard carbon electrodes depending on the type of material, the amount of conductive carbon and the method of homogenization. The article is well written and contains some interesting experimental results, however, strictly speaking, they are secondary. The most important characteristics of hard carbon-based electrodes - capacity and Coulombic efficiency - are not given by the authors, which does not allow us to speak about the high value of the results obtained. If the authors demonstrate that the discovered patterns are related to the electrochemical characteristics of hard carbon, this will significantly increase the scientific value of the work. Thus, I highly recommend presenting the results of galvanostatic cycling of sodium half-cells and discussing the data obtained with the findings already presented in the article.

Author Response

Thank you for your thoughtful consideration. As requested, we have conducted half-cell cycling experiments to show specific capacity and coulombic efficiency of electrodes using the hard carbon formula used in this work. 

Reviewer 2 Report

This manuscript reported a optimal processing and formulation of hard carbon electrodes for commercial sodium-ion cells. However, the manuscript should be improved by elucidating the following issues:

Q1. The SEM imaging data could be further improved. Fig 6c is not clear.

Q2. What's the loading and density of electrolyte? Authors should provide this information.

Q3. The full Na battery performance (using different electrodes) should be added. It can exhibit potential application of the hard carbon anode in this work.

Author Response

Thank you for your insightful suggestions. The image in Figure 6c has now been sharpened for clearer definition. The electrolyte was weighed at a set volume, and the density has been declared in the methods section. Description of electrolyte loading in cells has also been added; this was deliberately maximized by pre-wetting electrodes and separator to ensure incomplete surface wetting did not affect impedance values. While we do not have the facility to make full cells, we have made half-cells and cycled them galvanostatically to demonstrate this method using anodes matching those used in this work. 

Reviewer 3 Report

Recommendation: minor revision.

Comments: In this work, the authors assessed the effectiveness of blending carbon black into hard carbon anodes as a conductive additive for sodium-ion batteries (SIBs). This work provided a quick and reliable procedure to link changes in electrode formulation or processing with solid electrode conductivity, using fewer resources than traditional cell cycling. The experiment data relevant to the conductive networks in hard carbon anodes offered in this manuscript are sufficient to support the conclusion. So, I recommend that this manuscript can be accepted for publication in Coatings after minor revision.

1. The introduction of this paper needs to make a strong argument about the impact and novelty of the work further. So, the introduction should enrich some related SIBs anodes in this section. Adv. Sci., 2022, 9, 2200247; Adv. Energy Mater., 2017, 7, 1602898.

2. How about the active mass loading on the electrode?

3. The authors better offer the Raman spectra of these hard carbon anodes to identify the state of carbon.

4. To demonstrate the electrochemical performance of these hard carbon anodes, the electrochemical performance is better offered. 

Author Response

Thank you for your helpful insights. As suggested, we have added several references to articles about hard carbon anodes that have been published since our original draft of this manuscript. We have also added the active material loading for a range of electrodes used in this work, which you can find in the new Appendix B. This varied with conductive additive since increased carbon black displaces hard carbon in the slurry formula, and the high surface area of carbon black also increases the viscosity of slurries, changing the thickness of spread films. To satisfy interest in the Raman spectra of the commercial carbons purchased for this work, we have included links to two publications that contain this information, with a brief explanation. We have also made and cycled sodium ion half-cells to demonstrate galvanostatic sodiation and desodiation.